# Long-Chain Acyl-CoA Synthetases Promote Poplar Resistance to Abiotic Stress by Regulating Long-Chain Fatty Acid Biosynthesis

**DOI:** 10.3390/ijms23158401

**Published:** 2022-07-29

**Authors:** Hui Wei, Ali Movahedi, Yanyan Zhang, Soheila Aghaei-Dargiri, Guoyuan Liu, Sheng Zhu, Chunmei Yu, Yanhong Chen, Fei Zhong, Jian Zhang

**Affiliations:** 1Key Laboratory of Landscape Plant Genetics and Breeding, School of Life Sciences, Nantong University, Nantong 226019, China; 15850682752@163.com (H.W.); cjqm1989@126.com (G.L.); ychmei@ntu.edu.cn (C.Y.); chenyh@ntu.edu.cn (Y.C.); fzhong@ntu.edu.cn (F.Z.); 2Co-Innovation Center for Sustainable Forestry in Southern China, Key Laboratory of Forest Genetics & Biotechnology, Ministry of Education, College of Biology and the Environment, Nanjing Forestry University, Nanjing 210037, China; zyy190103128@njfu.edu.cn (Y.Z.); zhusheng0701@foxmail.com (S.Z.); 3College of Arts and Sciences, Arlington International University, Wilmington, DE 19804, USA; 4Department of Horticulture, Faculty of Agriculture and Natural Resources, University of Hormozgan, Bandar Abbas 47916193145, Iran; s.aghaei6418@gmail.com

**Keywords:** LACSs, FAs, poplar, LCFA, VLCFA

## Abstract

Long-chain acyl-CoA synthetases (LACSs) catalyze fatty acids (FAs) to form fatty acyl-CoA thioesters, which play essential roles in FA and lipid metabolisms and cuticle wax biosynthesis. Although LACSs from Arabidopsis have been intensively studied, the characterization and function of LACSs from poplar are unexplored. Here, 10 poplar *PtLACS* genes were identified from the poplar genome and distributed to eight chromosomes. A phylogenetic tree indicated that PtLACSs are sorted into six clades. Collinearity analysis and duplication events demonstrated that *PtLACSs* expand through segmental replication events and experience purifying selective pressure during the evolutionary process. Expression patterns revealed that *PtLACSs* have divergent expression changes in response to abiotic stress. Interaction proteins and GO analysis could enhance the understanding of putative interactions among protein and gene regulatory networks related to FA and lipid metabolisms. Cluster networks and long-chain FA (LCFA) and very long-chain FA (VLCFA) content analysis revealed the possible regulatory mechanism in response to drought and salt stresses in poplar. The present study provides valuable information for the functional identification of PtLACSs in response to abiotic stress metabolism in poplar.

## 1. Introduction

As the main structure and component of membrane lipids, fatty acids (FAs) are essential for maintaining membrane integrity and providing energy for various metabolic processes. The oxidation of FAs provides a large amount of acetyl-coenzyme A (CoA), which can participate in the tricarboxylic acid cycle (TCA) to provide energy. FAs, as important signal molecules of biofilm, participate in responses to adverse conditions [1,2]. The initial activation of FAs in different metabolic pathways requires the formation of acyl-CoA intermediates mediated by acyl-CoA synthetase (ACS) enzymes [3]. The formation of acyl-CoA intermediates is an ATP-dependent process and catalyzes the two-step reaction: (1) Free FA + ATP → fatty acyl-AMP + PPi; (2) fatty acyl-AMP + CoA-SH → fatty acyl-S-CoA + AMP [4]. In plants, lipid metabolism pathways are associated with the biosynthesis of sphingolipid, lecithin, diacylglycerols (DAGs), and triacylglycerols (TAGs), and lipid and lipid derivatives serve as precursors of plant cutin and cuticular wax [5]. FAs are classified into short-chain FAs (SCFAs), medium-chain FAs (MCFAs), and long-chain FAs (LCFAs) based on the length of the FA carbon chain. They are also divided into saturated FAs (SFAs) and unsaturated FAs (USFAs) according to the saturation of the FA carbon chain [6]. Previous studies showed that LCFAs and their derivatives play a vital role in enhancing plant resistance to drought and cold stresses and even to improving the quality of products [7]. The LCFAs and their derivatives are associated with the formation and fertility of pollen [8], the specific recognition between pollen and stigma [9], the polar transport of auxin [10], and the formation of the equatorial plate during mitosis [11]. USFAs can increase the fluidity of the membrane, which is very important for activating the enzymes on the membrane. In plants, the higher USFA contents in membrane lipids can result in the temperature of transition of membrane lipids, which increases plant resistance to cold stress [12]. Therefore, the degree of USFAs in plant cell membrane directly determines the fluidity of the membrane and affects the resistance of plants to abiotic and biotic stresses [13]. 

The plant cuticle, a lipid barrier on the aerial surface of plant tissues (roots, stems, leaves, and flowers), is composed of LCFA, very long-chain FA (VLCFA), and their derivatives [14,15]. Plant cuticles can protect against non-stomatal water loss and improve resistance to biotic and abiotic stresses [16]. The plant cuticle biosynthesis requires the participation of LCFA, VLCFA, and many enzymes [17]. In addition, LACS can participate in the biosynthesis of triacylglycerol (TAG) by regulating FAs to form acyl-CoA thioesters [18]. In the entire catalytic process, the AMP-binding domain in the LACS performs the catalytic function, which belongs to a highly conserved domain in the carboxyl-CoA ligase superfamily [19]. Some special LACSs are considered as transporters of FAs between different subcellular compartments [20]. The spatial distribution of LACSs within the cell is comprehensive because of their participation in various metabolic pathways [21].

In Arabidopsis, nine AtLACS members (AtLACS1–AtLACS9) are classified into the LACS family, most of which participate in FA-derived molecule pathways and are determined to be localized in different cellular organelles [5,22]. AtLACS1, AtLACS2, and AtLACS4 are localized in the ER and are associated with wax or cutin biosynthesis [17,20,23]. Among them, AtLACS1, having a close association with the generation of very long-chain fatty acyl-CoAs (VLCF-CoAs, >C20), plays a chief part in cuticular wax biosynthesis. In contrast, AtLACS4 has partial redundancy with AtLACS1 in the biosynthesis of VLCF-CoAs. In addition, AtLACS2 incorporates C16 and C18 LCF-CoAs, which also play an important role in cuticular wax biosynthesis [5,24]. LACS9 was reported to be localized in the outer membrane of the plastid and had associated with triacylglycerol synthesis by transferring the LCF-CoAs from the ER to the plastid [20]. Compared with a single *lacs1* mutant, *lacs1* and *lacs9* double mutants show no detectable differences in wax components, indicating that AtLACS9 is unrelated to wax biosynthesis [20]. In addition, the knockout of *AtLACS8*, another ER-specific isoform, in *lacs4* and *lacs9* double mutant leads to lethal disruption, suggesting that AtLACS8 functionally overlaps with AtLACS4 and AtLACS9 [20]. There are no distinguishable changes in seed oil-related phenotypes between wide-type (WT) Arabidopsis and the *lacs6* and *lacs7* mutants. Compared with WT, the *lacs6* and *lacs7* double mutants display a defect in seed oil mobilization. The result suggests that AtLACS6 has a functional overlapping with AtLACS7 in mediating the β-oxidation of FAs in peroxisome [25]. Whereas *AtLACS3* was reported to be specifically expressed in leaf epidermal cells where the cutin layer is produced and *AtLACS5* was identified to be only expressed in flowers, the overall function of AtLACS3 and AtLACS5 on acyl-lipid metabolism has yet to be clarified [3,26].

Besides studies on the function of AtLACS members, a few LACS from other plant species have been characterized. In *Ricinus communis*, RcLACS2 catalyzes ricinoleic acid-S-CoA preferentially and relates to castor oil biosynthesis [27]. TpLACSA of *Thalassiosira pseudonana* could preferentially activate polyunsaturated VLCFA to its CoA thioester, accumulating high TAG levels [28]. MdLACS2, with high sequence homology with AtLACS2, plays an essential role in catalyzing the formation of C16:0 LCFA-CoA in apples [29]. GhACS1 from cotton, an orthologue of AtLACS4, is involved in anther development [30]. GmLACS2 from soybean has been identified to localize on the peroxisome and mediate FA degradation during seed germination [31]. HaLACS1 and 2 of *Helianthus annuu*, an ortholog of AtLACS8 and 9, are closely associated with seed oil synthesis [32]. In *Brassica napus*, BnLACS2 has a preference for C16:0, C18:0, C18:1, and C22:1 and participates in seed oil production [33]. LuLACS8A of flax (*Linum usitatissimum*) showed a substrate preference for C18:3 α-linoleic acid and contributed to seed oil biosynthesis [34]. The previous studies showed that LACSs play essential roles in lipid metabolism, FA catabolism, and cuticular wax synthesis and exhibit a special preference for the substrate among different plant species.

Poplar is one of the most commonly valuable raw materials in modern industry and is vital in regulating environmental ecology. However, poplar constantly exposed to environmental stresses during their growth and development limits its potential for timber yield. The surface wax has significantly protective functions in drought, salt, and other adverse environmental factors. Although the characterization and function of LACS in diverse species have been widely discovered, it is still necessary to explore the role and evolutionary relationship in poplar. Here, we systematically analyzed ten putative *PtLACS* genes from the entire *Populus trichocarpa* genome and explored the evolutionary relationship of LACS members among advanced and lower plants. We also examined the chromosomal localization and collinearity of *LACS* genes. In addition, we analyzed the expression patterns of *PtLACS* genes in different tissues and their response to various treatments. We investigated protein interaction networks among the PtLACSs to understand the putative regulatory networks related to LCFA and VLCFA biosynthesis in poplar. To further understand the putative mechanism of PtLACS in response to abiotic stress, we evaluated LCFA and VLCFA contents during the abiotic stress. Our results provide important insights into the profiles and functions of PtLACSs and lay the foundation for discovering the regulatory networks associated with wax biosynthesis and the abiotic stress response in poplar.

## 2. Results

### 2.1. Identification and Classification of PtLACS Family Genes

The *Arabidopsis thaliana* and *Oryza sativa* LACS protein sequences were retrieved from the Arabidopsis and rice database. To determine the putative LACS proteins in *P. trichocarpa*, nine AtLACS proteins as a query with a threshold e-value of 1 × 10^−10^ and identity of 40% was performed to BLASTP analysis in the *P. trichocarpa* genome database. In addition, the AMP-binding domain (Pfam: PF00501) was downloaded from a hidden Markov model (HMM) profile HMMER software. To further identify the LACS gene family in *P. trichocarpa*, the HMMER software was subjected to search the poplar genome database with a threshold e-value of 1e-5. The 10 *PtLACS* genes were renamed *PtLACS1-7*, *PtLACS8-1*,*2*, and *PtLACS9-1*,*2* according their homologs with Arabidopsis and rice *LACS* genes and phylogenetic relationship with AtLACSs and OsLACSs (Appendix A). The multiple sequence analysis showed that these *PtLACS*, *AtLACS* and *OsLACS* genes have no distinguishable difference in length, from 1632 bp (*PtLACS1*) to 2196 bp (*PtLACS8-1* and *PtLACS8-2*) (Appendix A). The ExPASy analysis showed that predicted molecular weights (MWs) of PtLACS, AtLACS, and OsLACS proteins range from 61.30 kDa (PtLACS1) to 79.88 kDa (PtLACS8-1). Also, the predicted pI of LACS values varied from 5.52 (PtLACS1) to 8.11 (AtLACS6). In addition, the instability index of LACSs varied from 27.25 (PtLACS8-1) to 40.68 (AtLACS4), and most LACS proteins were forecasted to be stable except for AtLACS4. The grand average of hydropathicity (GRAVY) was less than zero, except for PtLACS9-1 and 2, indicating that LACS proteins are hydrophilic. Moreover, the subcellular localization of LACSs was predicted by Cell-PLoc, and the result revealed that all LACS signals exist in peroxisome (Appendix A).

### 2.2. Multiple Alignments and Phylogenetic Analysis of PtLACSs

The enzymes with acyl-CoA synthetase activity contain two highly conserved domains, the AMP-binding domain ([T/S][S/G]G[T/S][T/E]GNPKG) and a conserved arginine (R) (TGDX7GX3hX[DG]RX4HX4GX2hX2[EK]hE). The multiple alignments of PtLACSs revealed that two highly conserved domains are found in PtLACSs (Appendix A). To investigate the evolutionary relationships among LACSs, the total of LACS protein sequences from 17 species, including *Galdieria sulphuraria*, *Ectocarpus siliculosu*, *Selaginella moellendorffii*, *Ananas comosus*, *O. sativa*, *Musa acuminate*, *Triticum Urartu*, *Zea mays, Solanum lycopersicum*, *Cucumis sativus*, *Pyrus bretschneideri*, *Nicotiana tabacum*, *P. trichocarpa*, *Vitis vinifera*, *Prunus persica*, *A. thaliana*, and *Citrus sinensis* were aligned using the ClustalW software based on the default parameters. In addition, MEGA7 has been applied with a neighbor-joining (NJ) method to construct a phylogenetic tree among 17 specie of LACS proteins. It was found through the NJ method that there are six clades in the phylogenetic tree of higher plants, such as clades I-VI (Figure 1). The LACS proteins in lower plants were assigned to an unremarkable clade, such as *Galdieria sulphuraria* or *Ectocarpus siliculosus*. Among clades I-VI, the monocot and dicot LACSs were classified into different subclades, such as the monocot and dicot LACSs gathering together. Interestingly, the indistinguishable branching relationships between the LACS3, 4, and 5 proteins in clade III and LACS6 and 7 proteins in clade IV supported the naming principles for the PtLACS proteins. The evolutionary relationship among LACSs may provide important insights for identifying poplar LACSs.

### 2.3. Conserved Motif and Gene Structure Analysis of PtLACSs

It is known that the conserved protein sequence is considered the consensus motif [35]. The conserved motifs are putative components for the evolution and diversity of the gene family. The conserved motifs of LACSs from Arabidopsis, rice, and poplar were examined to explore the diversification of LACSs further. In total, 10 conserved motifs were analyzed in LACS members, motifs 1–10 were common in all clades, and only motif 3 was lacking in AtLACS1(Appendix A). Based on InterProScan and GenomeNet annotations, motif 5 comprised an AMP-binding domain (Appendix A). The specific amino acid residues ([T/S][S/G]G[T/S][T/E]GNPKG) within motif 5 were related to AMP-binding signatures, which may include participants in lipid formation and degradation. 

The gene intron-exon structure plays an important role in the phylogenetic relationship of gene family members. Therefore, the intron-exon organization for each *LACS* gene was examined to explore the structural diversity of *LACS* genes. The results displayed that the disclosed exon number of *LACS* genes varied from 11 to 23. Among the same LACS clade, although the lengths of *PtLACS* introns were much longer than those of *AtLACS* genes, the exon number and gene organization were relatively similar. For example, *PtLACS6* and *7* in clade IV contained 21 exons, consistent with *AtLACSs* and *OsLACSs* in clade IV. In addition, the untranslated regions (UTRs) showed in most *LACS* members except for *AtLACS5* and *PtLACS8-2* (Appendix A).

### 2.4. PtLACS 3D Structure and Phosphorylation Site Analysis

Protein structure is closely associated with protein function, and the disruption of protein structure would affect protein function. In order to further analyze the putative function of PtLACS proteins, the 3-dimensional (3D) structures of PtLACS proteins were constructed using SWISS-MODEL. Based on the analysis of AtLACS and PtLACS structures, the 3D models revealed that AtLACS and PtLACS proteins comprise alpha-helices, random coils, and beta strands (Appendix A). A similar structure was found in the same LACS clade which supported the sorting principles for the phylogenetic tree of LACS family members. In addition, the similar structure of LACS in the same clade implied that PtLACS proteins from the same clade might have a similar function during plant evolution (Appendix A).

The phosphorylation, a kind of post-translational modification, is usually involved in forming the mature protein, and the NetPhosK 3.0 Server usually predicts the kinase phosphorylation site. Based on the prediction of PtLACS phosphorylation, the serine site was identified as the most dominant phosphorylation site for PtLACS, followed by the threonine and tyrosine phosphorylation sites (Appendix A). Interestingly, the protein kinase C (PKC) was predicted to be the most likely phosphorylated protein in the various kinds of kinase phosphorylation proteins. PtLACS1-9 were predicted to have different phosphorylation modes. For example, besides the same phosphorylation mode with PtLACS1, PtLACS2 also applied the phosphorylation modes including CaM kinase II (CaM-II), epidermal growth factor receptor (EGFR), ribosomal S6 kinase (RSK) and sarcoma kinase (SRC). 

### 2.5. Chromosome Localization and Collinearity of PtLACS Genes Analysis

The chromosome (Chr) locations of *PtLACS* genes were mapped to the 8 Chrs of poplar. The physical locations showed that 10 *PtLACS* genes were unevenly distributed on poplar Chrs (Appendix A). Chr01, Chr03, Chr05, Chr09, Chr10, and Chr13, and each of the remaining 6 Chrs had one *PtLACS* gene. In addition, two *PtLACS* genes were evenly distributed on Chr02 and Chr13.

Among *PtLACS* members, the collinearity relationships of the *PtLACS* genes were analyzed to explore the genomic expansion mechanism. In total, six of the 10 *PtLACSs* had syntenic relationships, four *PtLACSs* in two gene pairs *(PtLACS9-1* and *PtLACS9-2*, and *PtLACS3* and *PtLACS4*) were segmentally duplicated and mapped to four different Chrs (Figure 2), and two *PtLACSs* in one syntenic pair (*PtLACS8-1* and *PtLACS8-2*) underwent tandem duplication. In conclusion, segmental duplication may be essential in expanding the *PtLACS* gene family. The non-synonymous (Ka) and synonymous (Ks) between syntenic pairs were calculated to identify the selective pressure. Ka/Ks > 1 means positive selection, Ka/Ks < 1 means purifying selection, and Ka/Ks = 1 means accelerated evolution with neutral selection. In this study, the Ka/Ks ratios of 2 syntenic pairs of the four segmental duplication events were <1, indicating that two gene pairs may eliminate harmful mutations and have undergone purifying selection. The Ks values of segmental duplication events (0.171 and 0.199) were significantly higher than tandem duplication events (0.060), which indicated that tandem duplication events occurred later than segmental duplication events.

Syntenic maps were performed to identify paralogous *LACS* gene families among poplar, Arabidopsis, rice, and willow to analyze gene duplication and the evolution of *LACS* genes among these species (Figure 3). According to the syntenic relationships, 11 syntenic pairs of paralogous genes were found between *P. trichocarpa* and *A. thaliana*, 11 gene pairs of paralogous genes were detected in *P. trichocarpa* and *Salix purpurea*, and only one pair of paralogous genes was found between *P. trichocarpa* and *O. sativa*. Interestingly, the 11 paralogous gene pairs included seven *PtLACSs* and seven *AtLACSs*, and each *PtLACS* gene displayed one to three paralogous *AtLACSs*. For example, both *PtLACS3* and *4* shared three syntenic pairs with *AtLACSs*. Also, the localization of the *PtLACSs* on the Chrs partially corresponded to that of willow. For example, most of the *PtLACSs* on Chr01, 02, 03, and 05 correspond to Chr01, 02, 03, and 05 of willow. *PtLACS3* and *4* were related to syntenic gene pairs (especially among poplar, Arabidopsis, and willow *LACS* genes), which suggested that syntenic pairs might play an essential role in the evolutionary process of the *LACS* gene family.

### 2.6. Cis-Acting Element Analysis of PtLACS Genes

The 2000 bp upstream sequences of the *PtLACS*, *AtLACS*, and *OsLACS* genes were extracted, and the *cis*-acting elements in the promoters of *PtLACS*, *AtLACS*, and *OsLACS* genes were predicted to elaborate the gene function. The most common *cis*-elements in the *PtLACS*, *AtLACS* and *OsLACS* promoters involved light responsiveness, such as G-box, TCCC-motif, and GATA-motif (Appendix A). Furthermore, the *cis*-elements of anaerobic induction, low-temperature, and hormone reactions were divided into *PtLACS*, *AtLACS*, and *OsLACS* promoters, indicating that the LACS family may have a close association with plant growth regulation and hormonal responses. The *cis*-elements were classified into three main clades: light response elements, plant growth, development-related elements, and stress responsive elements. The first clade was involved in light response elements, which maintain basic life in plants. The second clade comprised cis-elements associated with plant growth and development, such as endosperm expression, meristem expression, seed-specific regulation, and cell cycle regulation. The third clade comprised *cis*-elements that participated in various stress responses, such as ABA responsiveness (ABRE), gibberellin (GA) responsiveness, auxin (IAA) responsiveness, and methyl-jasmonate (Me-JA) responsiveness. It is known that MYB transcription factors are important regulators involved in plant physiological responses. The MYB binding sites of *PtLACS*, *AtLACS*, and *OsLACS* promoters were predicted to participate in drought, light responsiveness, and flavonoid biosynthesis, indicating that the *PtLACS*, *AtLACS*, and *OsLACS* genes are regulated by MYB transcription factors that play roles in physiological processes. Identifying *cis*-acting elements in the *LACS* promoters showed that LACSs might participate in multiple stress responses and respond to various physiological processes.

### 2.7. Interaction Network and GO Enrichment Analysis of PtLACSs

To explore the putative signaling pathway or physiological process of LACSs, a protein interaction network was performed to identify the AtLACS and PtLACS members using the String database (https://string-db.org/ (accessed on 10 May 2022 )). In the interaction network of AtLACSs, most proteins predicted to interact with AtLACSs were known. The interaction between AtLACS2 and AT2G45970, cytochrome P450, may be involved in the cutin and wax biosynthesis [15,36]. The interaction between AtLACS1 and AT5G04040, sugar-dependent1 (SDP1), encoding a patatin-domain triacylglycerol lipase, plays an important role in storage lipid breakdown during seed germination [37]. The interactions between AtLACS6 and the ACX1, 4, 5, and 6, acyl-CoA oxidases closely associate with LCFA biosynthesis [38]. The interaction between AtLACS6 and AT5G56290, encoding a peroxisomal targeting signal type 1 receptor, coordinates peroxisomal protein translocation [39]. The interaction between AtLACS6 and AT2G33150, encoding a peroxisomal 3-ketoacyl-coa thiolase 3, participants in FA beta-oxidation during germination and subsequent seedling growth [40] (Appendix A). Most proteins interacting with PtLACSs were not revealed in the poplar interaction network. Two exceptions were Potri.001G155500, peroxisomal acyl-CoA oxidase 2, and Potri.002G092700.1, and acetyl-CoA carboxylase. The results on the poplar interaction network suggested that PtLACSs and interaction proteins may be involved in lipid formation and degradation (Appendix A). The LACS family widely participates in FA biosynthesis according to the interaction network. To further explore multiple biological processes involved in LACSs, a GO annotation of *LACSs* and interaction-related genes were performed to identify the specific functions of the LACS and interaction-related proteins. In Arabidopsis, the AtLACSs and interaction-related proteins were significantly involved in molecular function, cellular component, and biological process (Figure 4A). Based on the GO enrichment analysis, the *AtLACSs* were enriched in acyl-CoA dehydrogenase activity, acyl-CoA oxidase activity, monocarboxylic acid catabolic process, cellular lipid catabolic process, fatty acid catabolic process, lipid oxidation, and fatty acid oxidation. It is suspected that AtLACSs may participate in fatty acid and lipid metabolisms by regulating the activities of AtLACSs and interaction-related proteins. In poplar, GO enrichment analysis showed *PtLACSs* and interaction-related genes are also associated with molecular function, the cellular component, and the biological process (Figure 4B). The biological process concentrated primarily on the FA and lipid metabolisms, suggesting that PtLACSs and interaction-related proteins may be essential in supplying FA and lipids for plant growth and stress response. The GO annotation offers a foundation for further study on the function of the *PtLACS* family gene and provides clues about the regulatory pathways involved in FA and lipid metabolisms.

### 2.8. Expression Patterns of PtLACSs in Various Tissues

To explore the roles of PtLACSs, the expression patterns of *PtLACSs* in various tissues of *P. trichocarpa*, ‘Nanlin 895’ and ‘Shanxinyang’ were identified using qRT-PCR. The current study showed that the transcript levels of *PtLACSs* differed in the leaf, stem, and root of *P. trichocarpa*. For example, *PtLACS7* and *8-1* had the highest transcript levels in mature leaf (ML), while *PtLACS3* was highly expressed in young leaf (YL). Furthermore, the relative expression levels of *PtLACS9-1* and *9-2* were higher in the root. In addition, *PtLACS1*, *2*, and *6* were expressed at relatively higher accumulations in an upper region of the stem (US) (Figure 5A). In ‘Nanlin 895’, five *PtLACS* genes (*PtLACS2*, *6*, *7*, *8-1*, and *9-1*) showed relatively lower expression levels in the root. Also, the relative transcript levels of *PtLACS4* and *9-1* were expressed at low levels in a lower region of the stem (LS). The relative expression accumulations of *PtLACS2* and *8-1* increased approximately 11- or four-fold in YL, respectively (Figure 5B). In ‘Shanxinyang’, the relatively higher transcript levels of *PtLACS1*, *2*, *3*, *9-1*, and *9-2* were examined in YL, while the *PtLACS8-1* were expressed at relatively low levels in YL. In addition, *PtLACS1*, *4*, *7*, and *9-2* expression in root was relatively lower (Figure 5C). The divergence of *PtLACS* genes in different tissues may imply that PtLACS members are involved in different physiological processes. The same *PtLACS* gene in different strains of poplar has a different expression pattern, which heralds the functional differentiation of PtLACSs during artificial selection.

### 2.9. Expression Patterns of PtLACSs in Response to Different Stress Conditions

To identify whether PtLACSs correspond to stress conditions in poplar, the expression levels of *PtLACSs* in ‘Nanlin 895’ under the different stresses were measured using qRT-PCR. Under the ABA treatment, the transcript accumulations of *PtLACS6* and *9-2* were improved, and the relative expression accumulations were close to 20-fold, while the expression level of *PtLACS8-1* was down-regulated with a time course of ABA treatment. Furthermore, the transcript levels of *PtLACS1* and *3* increased first, followed by a gradual decrease. For most *PtLACS* genes, the peaks of PtLACS expression levels were found at 6 or 12 h after ABA treatment. The most significant mRNA accumulation of *PtLACS9-2* was identified at 6 h after ABA treatment. In addition, *PtLACS1*, *2*, *3*, *4*, and *9-1* mRNA levels at 48 h were down-regulated relative to the untreated control (Figure 6A).

The mRNA levels of *PtLACS* genes were changed with a time course of NaCl treatment (Figure 6B). For example, the transcript levels of *PtLACS7* and *9-2* were improved with a time course of NaCl treatment. Also, the highest mRNA levels of *PtLACS7* and *9-2* were examined at 24 h, the peaks of *PtLACS3* and *4* expressions were identified at 1 h, and the highest transcript levels of *PtLACS8-1*, *9-1*, and *9-2* were exhibited at 24 h after NaCl treatment. Interestingly, the *PtLACS1* expression level was significantly down-regulated after NaCl treatment relative to the untreated control. *PtLACS3* expression changed significantly for 1 h and declined during 6–48h.

Under the H_2_O_2_ treatment, mRNA levels of *PtLACS1*, *2*, *3*, and *8-1*, except for the special time course, were down-regulated over a time course of H_2_O_2_ treatment (Figure 6C). *PtLACS9-1* and *9-2* transcript levels were improved under H_2_O_2_ treatment, and the mRNA levels of *PtLACS9-1* and *9-2* peaked at 48 h or 6 h of H_2_O_2_ treatment, respectively. Interestingly, the *PtLACS3* and *6* transcript levels showed a dominant increase in the early stages of H_2_O_2_ treatment, while *PtLACS1*, *2*, *7*, and *8-1* transcript levels exhibited a significant decrease in the early stages of H_2_O_2_ treatment. Under the PEG_6000_ treatment, the expression levels of *PtLACS3*, *4*, *6*, *7*, and *9-2* were significantly up-regulated within 1–48 h. However, the mRNA levels of *PtLACS2* and *9-1* maintained low transcript levels within the time course of the PEG_6000_ treatment (Figure 6D). The expression profiles of *PtLACSs* under the same stress were divergent, indicating that functional differentiation of PtLACSs occurred in the evolutionary process. The mRNA pattern of the same *PtLACS* under the different stresses was also divergent, speculating that each PtLACS family member may respond to a specific treatment.

### 2.10. Cluster Network Analysis among PtLACSs and PtLTPGs

The previous studies showed that LTPs are associated with cuticular wax formation [41], and type G non-specific lipid transfer proteins (LTPGs) are involved in amounts of primary alcohols and hydroxy FAs with C16 and C18 in Arabidopsis seed coats [37]. The cluster network was performed to explore the putative relationship between PtLACSs and PtLTPGs [38]. It is known that similar expression patterns of gene modules may be associated with the same biological processes [42]. The transcript levels of *PtLACSs* and *PtLTPGs* were identified to determine whether PtLACSs are involved in cutin biosynthesis or respond to abiotic stresses by regulating LCFAs and VLCFAs. Under the ABA treatment, the cluster network of *PtLACSs* and *PtLTPGs* mRNA profiles was classified into two clades (Figure 7A). *PtLACS9-2* and *PtLTPG23*, *PtLACS7* and *PtLTPG28*, *PtLACS6* and *PtLTPG10*, *18*, *PtLACS1* and *PtLTPG4*, *PtLACS2* and *PtLTPG12*, *PtLACS1*, *3*, *8-1* and *9-1* and *PtLTPG4* clustered together in the dendrogram, respectively. Cluster network analysis of the effect of the NaCl on *PtLACS* and *PtLTPG* expression levels dominantly showed that they are clustered into two clades. *PtLACS2* and *PtLTPG21*, *PtLACS7* and *PtLTPG26*, and *PtLACS8-1* and *PtLTPG23* clustered together in the network, indicating that they are associated with LCFA biosynthesis in response to NaCl stress. In addition, *PtLACS1*, *3*, and *4* and *PtLTPG4* and *13* were sorted into the same dendrogram, suggesting that PtLTPG4 and 13 play an essential role in transporting LCFAs and VLCFAs synthesized by PtLACS1, 3, and 4 to form cuticular wax (Figure 7B).

For the PEG_6000_ treatment, the clustering clade was also divided into two parts, one of which had *PtLACS3*, *4*, *6*, *7*, *8-1*, and *9-2*; the other contained *PtLACS1*, *2*, and *9-1*. *PtLACS1* and *9-1* clustered with *PtLTPG1* and *35*, and *PtLACS2* clustered with *PtLTPG12*, indicating a close relationship in response to PEG_6000_ treatment. Also, *PtLACS6* and *7* and *PtLTPG36* gathered together in the clustering clade, suggesting that *PtLACS6* and *7* are closely associated with *PtLTPG36* in modulating cuticular wax biosynthesis (Figure 7C). Under the H_2_O_2_ treatment, the expression profiles of *PtLACS1* and *2* were like those of *PtLTPG14* and *32*, which suggested that PtLACS1 and 2 and PtLTPG14 and 32 modulate the response to H_2_O_2_ treatment by regulating cuticular wax biosynthesis. The transcript level of *PtLACS7* was similar to *PtLTPG21* and *28*, which suggested that PtLACS7 and PtLTPG21 and 28 may have the similar function in response to H_2_O_2_ treatment stress (Figure 7D).

### 2.11. LCFAs and VLCFAs Analysis before and after Osmotic Stress

Previous studies have determined that LACSs play an essential role in converting LCFAs and VLCFAs into cuticular wax biosynthesis, which improves plant capacity to respond to various stresses [3]. To explore whether LCFAs and VLCFAs function in response to osmotic stress, the GC-MS was used to analyze LCFA and VLCFA contents in poplar before and after drought and salt treatments. Under the drought treatment, the contents of palmitoleate (C16:1), palmitate (C16:0), cis-10-heptadecenoate (C17:1), margarate (C17:0), linoleic acid (C18:2), octadecanoate (C18:0), eicosenoate (cis-11) (C20:1), eicosatrienoate (cis-8,11,14) (C20:3), trans-11-eicosenoate (C20:1T), icosanoate (C20:0), heneicosanoate (C21:0), and other VLCFA (C22-C31) were decreased significantly, while the oleate (C18:1), linolenic acid (C18:3), eicosadienoate (cis-11, 14) (C20:2), erucate (C22:1) and cis-4,7,10,13,16,19-docosapentaenoate (C22:5) contents were improved significantly by drought stress. For salt treatment, the contents of oleate (C18:1), linolenic acid (C18:3), eicosadienoate (cis-11, 14) (C20:2), and behenate (C22:0) were up-regulated significantly, while other LCFA and VLCFA contents were apparently down-regulated (Figure 8). The above results indicated that LCFA and VLCFA contents are involved in osmotic stress and function in osmotic stress by regulating LCFA and VLCFA biosynthesis. In addition, the poplar was treated with osmotic stress after 15 d, and the poplar suffered great damage. Although the LCFAs and VLCFAs participate in cuticular wax, which can improve the water retention capacity, the poplar prefers to improve the fluidity of membrane lipids by increasing linolenic acid content (Figure 9).

## 3. Discussion

LACS can catalyze LCFA precursors to form into fatty acyl-CoA thioesters, which is essential in the biosynthesis of plant cuticles [16]. The cuticle is mainly composed of cutin and wax as an important protective substance. Cutin consists of glycerol and C16 and C18 FA derivatives; wax is alkane, alcohol, or ketone and is mainly composed of VLCFAs and corresponding derivatives [29,43]. Therefore, LACS plays a critical role in the biosynthesis of cutin and wax and participants in plant development and stress response. The biosynthesis of cutin goes through two processes, which are produced in two different organelles. The LCF-acyl-CoAs occur in the plastid, and the modification and final synthesis of VLCFA and corresponding derivatives generate in ER. Wax biosynthesis originates from de novo FA starts in the plastid, and then LCF-acyl-CoAs are transferred to ER for elongation and modification [43]. The characterizations and profiles of LACSs from different species were well identified. The research programs proved that LACSs have close associations with FA metabolism [3], i.e., nine *LACS* family genes were studied in the Arabidopsis and rice [22,44]. Although *LACS* genes are involved in the biosynthesis of cuticles and respond to various stresses in plants, the information and functional analysis of LACSs has not been reported in poplar, an economic woody plant cultivated worldwide. In addition, 34 *LACS* genes were determined in *Brassica napus*, 11 *LACS* genes were found in *M. domestica*, and 11 *LACS* orthologs were present in Arabidopsis [3]. In this study, 10 poplar *LACS* genes were identified in a poplar genome. The phylogenetic tree showed that the LACSs widely occurred in many species, including the lower and higher plants, and the LACSs in high plants could be classified into six clades. In comparison, the LACSs in lower plants had no remarkable classification. The cluster network, GO annotation, and contents of LCFA and VLCFA analysis implied that PtLACSs play an essential role in LCFA and VLCFA biosynthesis and are associated with stress resistance by regulating the cuticle, laying the foundation for further poplar research.

The phylogenetic tree of LACS members was performed to understand evolutionary relationships across plant species better. Based on the multiple alignments on LACS protein sequences, the phylogenetic analysis suggested that the LACSs were classified into two main branches and an additional branch of lower species. The divergence between the higher plant and lower species (algal) LACS might have originated from biological function differentiation. For example, the algal LACS was predicted to function in chloroplast and preferred chloroplast membrane FA biosynthesis, while the higher plant LACS localized in plasmid and ER for degradation and synthesis of LCFA and VLCFA. On two main branches of advanced plant LACS family members, one branch consisted of clades I, II, III, and IV, and the other branch were composed of clades V and VI. The genetic and functional diversity of evolutionary relationships portended two different evolutionary directions for LACS proteins in monocots or dicots. Clades II and III shared the exact origin and had the highest affinity [4,24]. The sequence alignments showed that the AMP-binding domain and conserved arginine (R) domain in PtLACS and AtLACS members are highly conserved, regardless of the LACS member clade. In addition, based on the conserved motifs and gene structures of *LACS* members, those LACS proteins in the same clade shared a similar motif composition and exon/intron distribution, but PtLACS1 had divergence in motif distribution with AtLACS and OsLACS. Indeed, LACS proteins clustered into the same clade typically have similar motif distribution and exon/intron structure, implying that LACSs in the same phylogenetic clade may be involved in a similar biological process. However, finding the diversity in the number, length, and positions of introns was interesting. Therefore, we speculated that LACS intron gain or loss events that occurred in the evolutionary process could cause the functional variations of LACS members. Previous studies discovered that small or large scale/whole-genome duplication events resulting in genome duplication could lead to the differentiation of the species [19], and tandem and segmental duplications were the main reason for producing gene family [45]. Therefore, the tandem and segmental duplications in poplar LACS gene family were identified, and a total 2 gene pairs for segmental duplication and one gene pair for tandem duplication were determined in all *PtLACSs*, indicating that *PtLACS3*, *4*, *8*, and *9* provide a primary source for *PtLACS* family expansion and play a critical role in poplar functional variations of PtLACSs.

Protein interaction network analysis is an excellent tool for investigating candidate proteins associated with standard-related biological processes [46]. In this study, the PtLACS and AtLACS protein interaction prediction was performed to analyze the putative profiles and functions of LACS members. ACX, a member of the acyl-CoA oxidase family, is deposited and controls the formation of LCFAs and corresponding derivatives. PTS1, encoding a peroxisomal targeting signal type 1 receptor, plays a critical part in seeding growth and development by modulating the process of FA beta-oxidation. ACC, acetyl-CoA carboxylase, catalyzes acetyl-CoA to form malonyl-COA, which plays an essential role in glucose and lipid metabolisms. Previous studies have determined that the transcriptional regulation of wax metabolism improves wax accumulation, and the accumulation of cuticles could contribute to plant tolerance to water deprivation [47]. The results suggested that LACSs may play an important role in abiotic stress response by the regulatory effect of these genes involved in the wax synthesis. The GO annotation was used to further investigate the functions of PtLACSs in wax biosynthesis and the regulatory pathway associated with PtLACSs and related interaction proteins. The mutation of *atlacs2* drastically reduced the dicarboxylate cutin monomer, which resulted in a principal reduction of components of Arabidopsis cutin [48]. Compared with the mutation of *atlacs2*, the mutation of *atlacs1* and *atlacs2* led to more severe cuticle defects, signifying functional overlap between AtLACS1 and AtLACS2 [3]. In addition, AtLACS6 and 7 participated in lipid degradation and other energy-requiring biological processes [22]. Based on the GO enrichment analysis in this study, PtLACSs were likely involved in the molecular function, the cellular component, and the biological process. The molecular function mainly concerned the acyl-CoA oxidase activity and acyl-CoA dehydrogenase activity, implying that PtLACSs play important parts in the oxidation and dehydrogenation of acyl-CoA active metabolic intermediates in FA synthesis and decomposition. The biological process mainly contains FA oxidation, lipid oxidation, FA catabolic process, FA beta-oxidation, monocarboxylic acid catabolic process, and cellular lipid catabolic process, which suggests that PtLACSs play a crucial role in FA and lipid biosynthesis and the catabolic process.

In Arabidopsis, *AtLACS* genes were highly expressed in various tissues and organs, including the leaf, stem, root, and flower [3]. Also, *AtLACS* genes had tissue-specific expression profiles. For example, *AtLACS1-3* and *6* were specifically exhibited in epidermal cells of the leaf where cuticular wax is generated; the higher expression levels of *AtLACS2-3* and *AtLACS9* were identified in endodermal cells of the root where suberin is produced. In addition, 18 of 34 *BnLACS* genes from *B. napus* were expressed in developing seeds. Some *BnLACS* genes displayed diversified expression profiles, while others displayed tissue-specific expression profiles. For example, the *BnLACS5s*, orthologous with *AtLACS5*, were specifically found in the bud, stamen, and anther [49]. *BnaLACS4*, the *AtLACS4* homologous gene in *B. napus*, was demonstrated to be localized in ER and involved in lipid metabolism [50]. *MdLACS* genes from apple had higher expression levels in pericarp tissue where wax and cutin are synthesized [4] The transcript levels of *PtLACS* genes were determined in various tissues of different poplar strains, and the result revealed that *PtLACS* genes were expressed in tissues tested, including ML, YL, US, LS, and root. The mRNA levels of *PtLACS9s* significantly accumulated in the root of *P. trichocarpa*, while the expression levels of *PtLACS7* and *PtLACS8-1* were dominantly illustrated in ML. The *PtLACS3* and *9-1* were highly expressed in the root of ‘Nanlin 895’, while the relatively higher *PtLACS4* and *8-1* expression levels were determined in YL. The transcript levels of *PtLACS1-3*, and *9s* significantly accumulated in YL, while the higher expression of *PtLACS4*, *7*, and *8-1* were illustrated in ML. These results suggested that divergent *PtLACS* expression patterns occurred in tissues of different poplar strains. We speculated that PtLACS members might have a functional differentiation in the FA or lipid metabolism process. 

It is well identified that LACSs are associated with stress resistance by regulating LCFAs or VLCFAs for cuticle biosynthesis [3]. Also, there are many studies where *LACSs* expression is connected with abiotic stress, including drought, salt, and phytohormones [4]. For instance, apple *MdLACS1*, *2*, *6*, *8*, and *9* expression levels significantly induced drought treatment. In addition, the expression levels of *MdLACSs* were investigated during salt stress, and salt treatment caused a significant upregulation of *MdLACS1* and *9* transcript levels [4]. Here, our results were: the expression levels of *PtLACS7* and *9-2* were highly induced under NaCl treatment; the *PtLACS6* and *9-2* were dominantly promoted during ABA treatment; the mRNA levels of *PtLACS7* and *9s* were significantly improved when the poplars were treated by H_2_O_2_ treatment; and the transcript levels of *PtLACS3*, *4*, *6*, *7* and *9-2* were up-regulated remarkably under the PEG stress. These results indicated that PtLACSs play crucial roles in responding to poplar abiotic stress. Still, to our present evidence, there is no putative mechanism of PtLACSs regulating abiotic stress in poplar. Integration of the fact that LTPGs transport lipids for cuticular wax and the LACSs function in LCFAs and VLCFAs for cuticle biosynthesis, we proposed that PtLACSs can regulate cuticle that forms surface barriers against biotic stress. The clustering was performed on expression patterns of *PtLACSs* and *PtLTPGs* before and after abiotic stress. The *PtLACS7* and *PtLTPG26* gathered together in the cluster dendrogram, illustrating that PtLACS7 and PtLTPG26 are associated with LCFA and VLCFA biosynthesis and transport in response to NaCl treatment. The *PtLACS9-2* was clustered with *PtLTPG9*, *26*, and *28*, which implied that PtLACS9-2 has a close association with PtLTPG9, 26, and 28 and may be involved in response to NaCl treatment through regulating LCFA and VLCFA biosynthesis. Therefore, based on the cluster dendrogram, we speculated that PtLACSs could guide LCFA and VLCFA biosynthesis and modulate LCFA and VLCFA contents in response to osmotic stress. The integration of contents of LCFA and VLCFA and the cluster dendrogram of *PtLACSs* and *PtLTPGs*, the putative working model of PtLACSs involved in osmotic stress responses, is proposed. In addition, the poplar was damaged in the early stage, the transcript levels of *PtLACSs* were promoted, and the activities of PtLACSs were increased. The improvement of LCFA and VLCFA contents accelerated the cuticular wax’s function in water retention and abiotic stress. However, when the poplar suffered extensive damage in the later stage of abiotic stress, the contents of LCFA and VLCFA decreased significantly. The ratio of linolenic acid and linoleic acid was increased dominantly, implying that promotion of USFA contents, such as linolenic acid content and linolenic acid/linoleic acid ratio, is the main method for poplar resistance to abiotic stress.

## 4. Materials and Methods

### 4.1. Plant Materials and Abiotic Stress Treatments

The woody plant medium (WPM) (pH 5.8) supplement with 0.1 mg/L indoles butyric acid (IBA) was used to propagate *P. trichocarpa* plants. The Murashige and Skoog (MS) medium (pH 5.8) containing 0.3 mg/L IBA was applied to cultivate ‘Shanxinyang’ (*P. davidiana × P. bolleana Loucne*), and 1/2 MS medium was used to cultivate ‘Nanlin 895’ (*P. deltoides × P. euramericana*). The poplars were cultivated in a greenhouse at 23 °C and at 74% humidity. The ‘Nanlin 895’ seedlings were cultured in a 1/2 MS medium for seven days and then transferred to a 1/2 MS medium containing 0.2 M abscisic acid (ABA), 0.2 M NaCl, 2 mM H_2_O_2_, and 10% PEG_6000_ for abiotic stress. For the drought treatment of soil-grown poplars, irrigation for ‘Nanlin 895’ seedlings cultivated in soil for three months was suspended, and leaves were harvested after two weeks. The ’Nanlin 895’ seedlings were irrigated with 0.2 M NaCl for two weeks for salt treatment. All samples were immediately frozen using liquid nitrogen and stored at −80 °C in a refrigerator. Three experimental repetitions, each of six lines, were carried out.

### 4.2. Identification and Classification of LACS Proteins in Poplar

All files associated with sequence data for *P. trichocarpa*, *A. thaliana* and *O. sativa* were downloaded from Phytozome (https://phytozome-next.jgi.doe.gov (accessed on 10 May 2022 )). Sequences related to the nine AtLACSs were achieved from the Arabidopsis genome database (https://www.Arabidopsis.org/ (accessed on 10 May 2022)). The AtLACS proteins as a query were performed for BLASTP analysis in the *P. trichocarpa* genome database. The PtLACS candidates were integrated by HMMER software in poplar with the AMP-binding domain (PF00501). All candidate sequences were input into the Conserved Domain Database (CDD) (https://www.ncbi.nlm.nih.gov/cdd (accessed on 10 May 2022)) to determine that the protein domain PtLACS members were confirmed after removing incomplete sequences. 

### 4.3. Characteristics and Phylogenetic Tree of LACS Members

The AtLACS, OsLACS, and PtLACS members were submitted to ExPASy (http://web.expasy.org/protparam/ (accessed on 10 May 2022)) to identify the MW, theoretical PI, instability index, stability, aliphatic index, and GRAVY. All PtLACS proteins were submitted to predict serine, threonine, or tyrosine phosphorylation sites using NetPhos (https://services.healthtech.dtu.dk/service.php?NetPhos-3.1 (accessed on 10 May 2022)). The PtLACS three-dimensional (3D) structures were modeled and visualized by SWISS-MODEL (https://swissmodel.expasy.org (accessed on 10 May 2022)) and chimera software, respectively, and subcellular localization of PtLACSs was conducted by Cell-PLoc 2.0 (http://www.csbio.sjtu.edu.cn/bioinf/Cell-PLoc-2/ (accessed on 10 May 2022)). The MEGA7 software was used to construct the phylogenetic tree among PtLACS, AtLACS, and OsLACS proteins. The *PtLACS* genes were renamed based on the evolutionary relationship with the *AtLACS* and *OsLACS* genes (Appendix A). The LACS protein sequences from 17 species were aligned and established phylogenetic trees to further analyze the evolutionary relationship between advanced and lower plants.

### 4.4. Gene Structure, Conserved Motif, and 3D Structure Analysis of PtLACS Members

The poplar genome annotation file (*P. trichocarpa*_444_v3.1.gene.gff3) was used to identify the gene structures (intron-exon) of PtLACS members, and TBtools was performed to visualize the *PtLACS* gene structures. To analyze the conserved motifs, amino acid sequences of PtLACSs were submitted to MEME (http://meme-suite.org/tools/meme (accessed on 10 May 2022)). The gene structures and conserved motifs of AtLACS and OsLACS were analyzed using the above methods. The results on gene structures and conserved motifs were rearranged depending on the phylogenetic tree of *P. trichocarpa*, *A. thaliana*, and *O. sativa*. To further analyze the structures and functions of PtLACS proteins, the 3D models of PtLACS proteins were predicted and visualized using the SWISS-MODEL and Chimera software.

### 4.5. Chromosomal Location and Collinearity Analysis of PtLACS Members

The poplar genome annotation file (*P. trichocarpa*_444_v3.1.gene.gff3) contained the positional information of all *PtLACS* genes, and the gff3 file was used to identify the physical positions of *PtLACS* genes mapped to the chromosomes of *P. trichocarpa*. The collinearity of *PtLACS* genes in the *P. trichocarpa* genome and the syntenic orthologous relationship of *LACS* genes among *P. trichocarpa*, *A. thaliana*, *O. sativa*, and *S. purpurea* were identified and presented using the multiple collinearity scan toolkit (MCScanX). The segmental and tandem duplications in *P. trichocarpa* genome were obtained according to the collinearity relationship of *PtLACS* genes. MCScanX was used to calculate the value of Ka, Ks, and Ka/Ks using TBtools with a simple Ka/Ks calculator.

### 4.6. Cis-Regulatory Element, Interaction Network, and GO Analysis of PtLACS Members

After extracting the 2 kb upstream sequence of *PtLACS*, *AtLACS* and *OsLACS* genes, and the 2 kb upstream sequences were submitted to PlantCARE (http://bioinformatics.psb.ugent.be/webtools/plantcare/html/ (accessed on 10 May 2022)) to predict the cis-acting elements. In addition, the PtLACS and AtLACS protein sequences were submitted to the String database (https://string-db.org/ (accessed on 10 May 2022)) to identify the putative interaction networks. Cytoscape software was used to visualize the interaction networks of PtLACS and AtLACS proteins. The go-basic files on *A. thaliana* and *P. trichocarpa* annotation were selected for gene ontology (GO) analysis. TBtools was used to visualize GO annotation analysis.

### 4.7. RNA Isolation and Expression Patterns of PtLACS Genes

The samples were manually collected from poplar leaf, stem, and root and immediately flash frozen and stored in a freezer for RNA extraction. The CTAB method, as previously described [51], was applied to carry out RNA extraction. The 2% agarose gel electrophoresis and a NanoDrop One/OneC spectrophotometer (Thermo Scientific, Waltham, MA, USA) were used to determine the integrity and concentration of RNA, respectively. Total RNA (1 μg) was reverse-transcribed to the first-strand cDNA using the PrimeScript™ RT Master Mix (TaKaRa, Dalian, China). The gene specific primers (Appendix A) were designed and chosen to analyze the expression levels of *PtLACS* genes. The qRT-PCR assays were performed using the UltraSYBR Green I Mixture (CWBIO, Beijing, China) and the ABI 7500 Fast Real-Time PCR System (Applied Biosystems, Waltham, MA, USA). The PCR was performed with the following procedure: thermal cycling, including 95 °C for 5 min; 40 cycles of 95 °C for 10 s, 60 °C for 30 s, and 72 °C for 30 s; and dissociation-curve analysis, including 60 °C for 60 s and 95 °C for 15 s.

### 4.8. Gene Cluster Network Analysis of PtLACS and PtLTPG Genes

The previous studies exhibited Arabidopsis LTPGs that have distinct roles in LCFAs, VLCFAs and their derivatives, and that are participants in the synthesis or deposition of cuticular waxes [52,53]. In addition, Wei et al. [54] showed that PtLTPGs could promote the contents of LCFAs and have a close association with a waxy layer in poplar. To explore the putative relationship between *PtLACSs* and FA-transported genes (*PtLTPGs*), the weighted gene cluster network analysis (WGCNA) package in the R programming language was applied to identify gene cluster networks [55].

### 4.9. LCFAs and VLCFAs Analysis

The leaves of poplar treated by two weeks of drought and salt stresses were freeze-dried, and gas chromatography-mass spectrometry (GC-MS) was used to analyze the LCFA and VLCFA contents. All above samples were immersed in 3 mL of reaction solvent containing 1 mL of n-hexane and 2 mL of 10% acetyl chloride-methanol for 2 h at 95 °C. Next, 6 mL of 6% potassium carbonate solution was added to the mixture. The n-hexane obtained from the above mixture was then removed by vacuum concentration. The isolated solvents were injected into a GC-flame ionization detector (Thermo Trace1300, Waltham, MA, USA) or GC-MS (Thermo ISQ7000, Waltham, MA, USA).

The column temperature was kept at 140 °C for 5min and then increased to 180 °C at 10 °C/min. Subsequently, it was raised to 210 °C at 4 °C/min and then increased to 310 °C for 30min at 10 °C/min. The flame ionization detector was applied to analyze the LCFA and VLCFA contents. Chromeleon7.0 software was applied to analyze individual LCFA or VLCFA, and the NIST 17 (https://www.sisweb.com/software/ms/nist.htm database (accessed on 10 May 2022)) was used to classify LCFA or VLCFA data.

## 5. Conclusions

The cuticular wax forms surface barriers that respond to abiotic stress and several LACSs have been documented to have close associations with cuticle formation. In this study, 10 PtLACSs were determined from poplar, which was mapped on 8 Chros, and PtLACSs were clustered into six clades (clades I-VI). All members were predicted to localize in the peroxisome and had the AMP-binding domain. The analysis of *cis*-acting elements showed that PtLACSs might be involved in light, stress, and hormone-responsive processes. Interaction proteins and GO annotation analysis suggested that PtLACSs play essential roles in FA and lipid biosynthesis. The cluster dendrogram and LCFA and VLCFA contents further indicated that PtLACSs modulate cuticular wax biosynthesis by regulating the LCFAs, VLCFA, and deviants, which play a dominant role in water retention and drought and salt resistance. Although the regulatory mechanisms of PtLACSs resistance to drought and salt resistance were not ultimately revealed in the present study, these results will provide a theoretical basis for further study of the profiles and functions of PtLACSs.

## Figures and Tables

**Figure 1 ijms-23-08401-f001:**
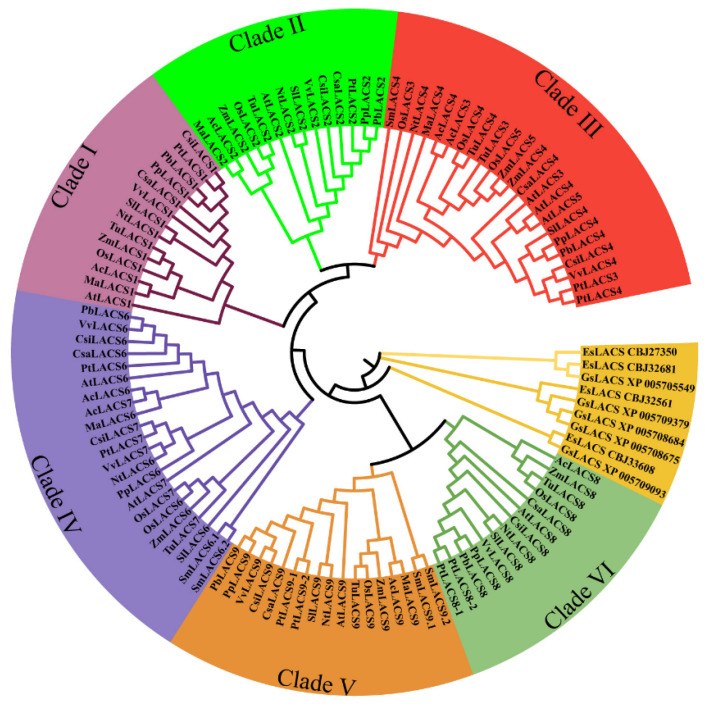
Phylogenetic tree and classification analysis of LACS members. The LACS protein sequences from different species are aligned using ClustalX2, and a phylogenetic tree is constructed using the neighbor-joining (NJ) method in MEGA7.0. The LACS members are classified into seven clades: I, II, III, IV, V, VI, and an unremarkable clade, and seven clades are highlighted in different colors.

**Figure 2 ijms-23-08401-f002:**
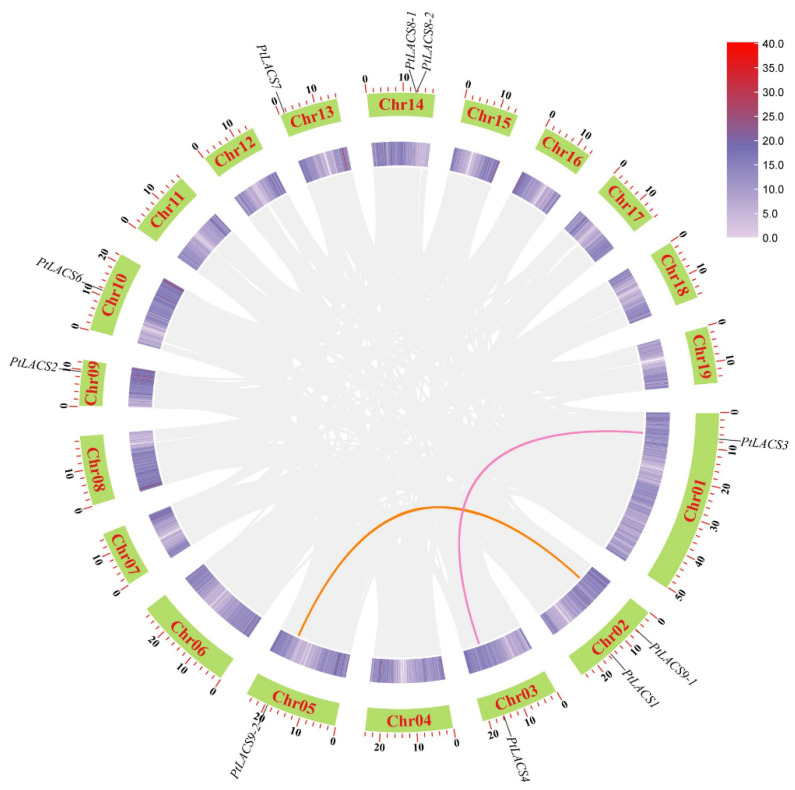
Synteny analysis of *PtLACSs* in poplar genome. The green rectangles are used to highlight Chros 01–19, and the dark- and light-purple lines represent gene density. The orange and light red are indicated collinearity pairs among *PtLACS* genes.

**Figure 3 ijms-23-08401-f003:**
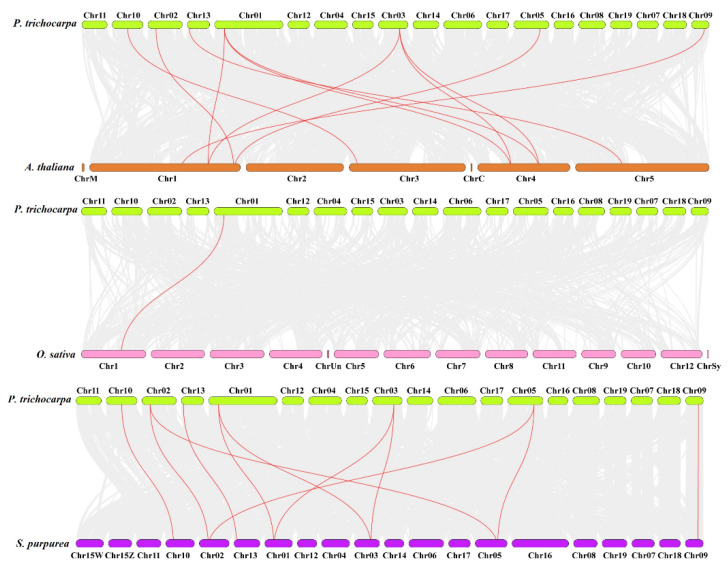
Collinearity analysis of the *LACSs* among poplar, Arabidopsis, rice, and willow. Gray lines represent orthologous gene pairs, and the red lines indicate the *LACS* gene pairs within poplar, Arabidopsis, rice, and willow.

**Figure 4 ijms-23-08401-f004:**
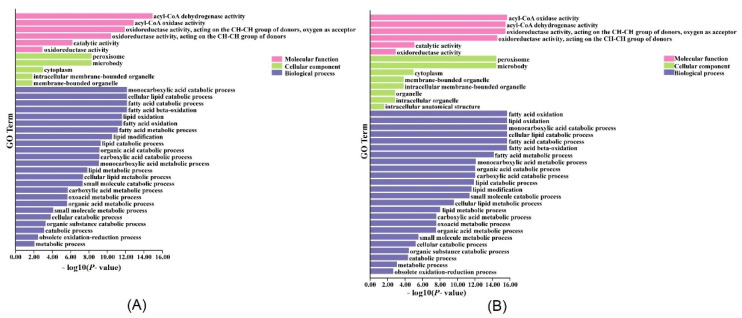
GO annotation analysis of *LACS* and interaction protein genes from Arabidopsis (**A**) and poplar (**B**) scattered in molecular function, cellular component, and biological process.

**Figure 5 ijms-23-08401-f005:**
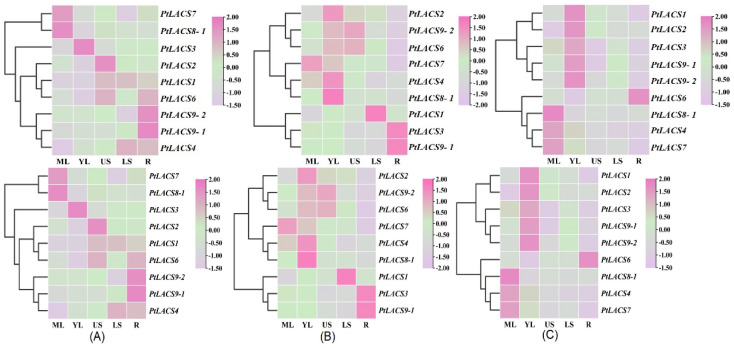
Relative expression analysis of the *PtLACS* genes in various tissues and oranges. The qRT-PCR was used to identify *PtLACS* gene expression patterns in *P. trichocarpa* (**A**), ‘Nanlin 895’ (*Populus × euramericana* cv.) (**B**), and ‘Shanxinyang’ (*P. davidiana × P. bolleana*) (**C**) tissues. The color scale indicates the normalized data, where pink represents a high expression level, lilac represents a low expression level, and light green represents a medium level.

**Figure 6 ijms-23-08401-f006:**
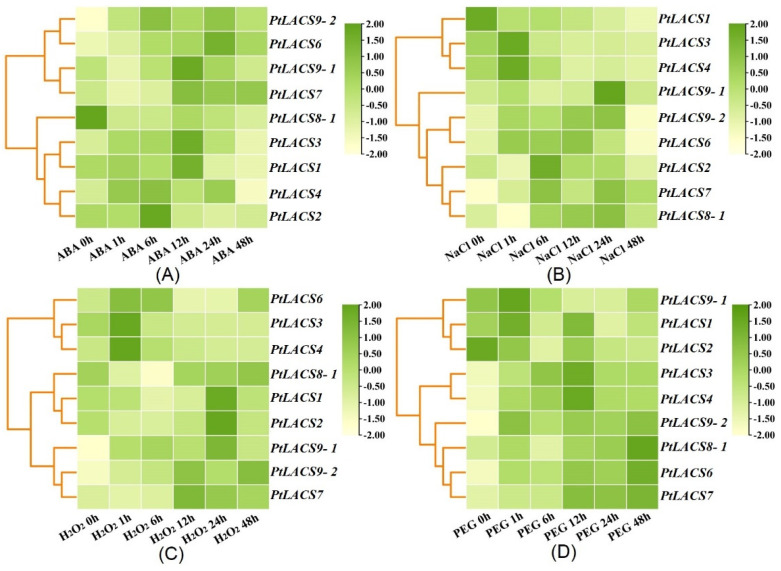
Relative expression analysis of *PtLACS* genes under the various stress treatments (**A**–**D**). The qRT-PCR was used to identify *PtLACS* gene expression levels under the ABA (**A**), NaCl (**B**), PEG_6000_ (**C**), and H_2_O_2_ (**D**). The color scale indicates the normalized data, where dark green represents a high expression level, light green represents a low expression level, and light yellow represents a medium level.

**Figure 7 ijms-23-08401-f007:**
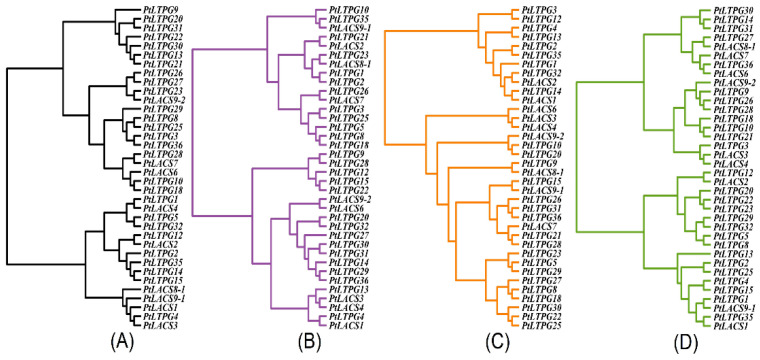
The cluster analysis of putative relationships between *PtLACS* and *PtLTPG* genes. The expression patterns of *PtLACS* and *PtLTPG* genes under the ABA (**A**), NaCl (**B**), PEG_6000_ (**C**), and H_2_O_2_ (**D**) are identified by cluster analysis of hclust by R language.

**Figure 8 ijms-23-08401-f008:**
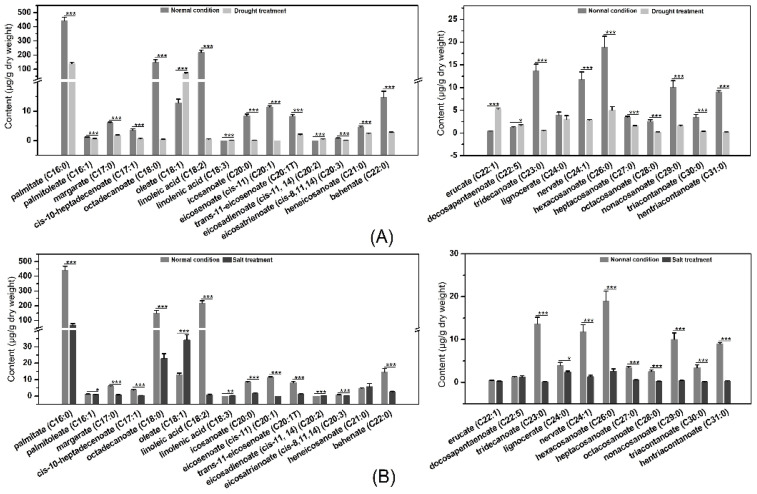
The analysis of LCFA and VLCFA contents under control and drought or salt stress conditions. (**A**) The LCFA and VLCFA contents in poplar grown under normal and drought stress conditions. (**B**) FA contents in poplar grown under normal and salt stress conditions. Three biological replicates were performed per group. Student’s *t*-test, *** *p* < 0.001, ** *p* < 0.01, and * *p* < 0.05 relative to control poplar.

**Figure 9 ijms-23-08401-f009:**
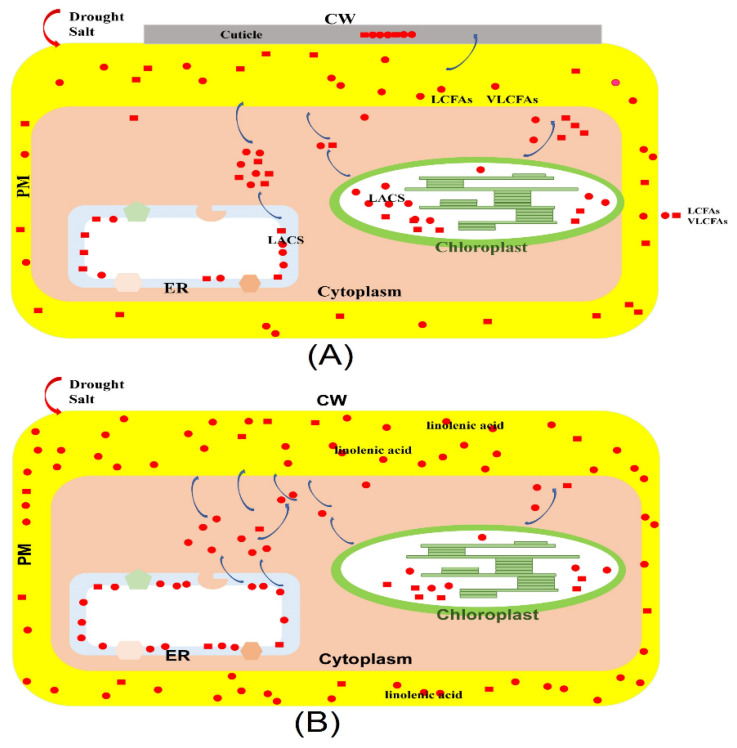
The putative working model of PtLACSs response to drought and salt stresses in poplar. Drought and salt stresses induce *PtLACS* expression accumulations. (**A**) In the early stage of drought and salt stresses, the improvement of *PtLACS* expression levels leads to the accumulation of the contents of LCFA and VLCFA, and LCFAs and VLCFAs contribute to the cuticular wax biosynthesis and promote the water retention capacity of poplar. (**B**) In the later stage of drought and salt stresses, the improvement of linolenic acid content increases the fluidity of the membrane and changes osmotic pressure.

## Data Availability

Not applicable.

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
