# Peer review of "Long-Chain Acyl-CoA Synthetases Promote Poplar Resistance to Abiotic Stress by Regulating Long-Chain Fatty Acid Biosynthesis"

_ijms, 2022, doi:10.3390/ijms23158401_

Round 1

Reviewer 1 Report

In the parts of world where environmental pollution decreased to the levels that allowed re-forestation, it is rather common to find a tree seemingly immune to biotic or abiotic stress among the group of less happy neighbours; yet the forestry scientist are unable to determine the exact cause of its endurance. The answer lies, very probably, on metabolic level, and deeper understanding to processes that influence the ability of trees to survive stress conditions is of utmost importance. Wax layer on the surface of leaves and needles is known to play an important role. It was already proved that the content of essential oils and other bioactive compounds in needles and leaves varies greatly even within one branch according to the age of needles and the position on the tree (i.e. due to the amount of exposition to direct light).

The paper "Long-chain acyl-CoA synthetases promote poplar resistance to 2 abiotic stress by regulating long-chain fatty acid biosynthesis" by Hui Wei, Ali Movahedi, Yanyan Zhang, Soheila Aghaei-Dargiri, Guoyuan Liu, Sheng Zhu, Chunmei Yu, Yanhong Chen, Fei Zhong and Jian Zhang uncovers part of this mystery, identifying 10 poplar genes responsible for the production of fatty acids that form fatty acyl-CoA thioesters, which play essential roles in FA and lipid metabolisms and cuticle wax layer creation. 

The paper shortly summarizes current state of knowledge, emphasizes the importance of understanding abiotic stress in poplars, then presents original findings and their interpretation. Description is given in good, simple scientific English, but the topic is rather complicated and more schematic charts would be welcome here and there. On the other hand, the paper already has 9 important figures and other reviewers might consider it too much, so minor revision won't be required because of this.

The authors could explain a bit more why they selected Populus trichocarpa. They mention environmental aspects, but P. trichocarpa is known to have fragile branches and is not exactly suitable as a roadside windbreaker. Was there a variety / cultivar? I.e. P. lasiocarpa is known to have a generic variety and endemic var. longiamenta.

Also, in the IJMS format where methods are described at the end of the paper, more information about samples and methods could be integrated into the results. The reader is interested in how and from what the results came (not in detail, few words), and usually is used to different format where this info is given above.

Suggestion: all your findings are based on PCR and gene identification. Do you have access to ambient imaging techniques (DART, DESI etc.)? In your case, interesting changes might occur in stressed samples as for the position of some metabolites of interest (higher concentration closer to the surface, closer to leaf edges etc.) Worth a try to look at it in your next paper.

Thanks for an interesting paper, it will be recommended to be accepted and it will be beneficial for the readers of IJMS.

Author Response

Dear Reviewer:

Thank you for the comments concerning our manuscript. The comments are all valuable and helpful for revising and improving our paper and the important guiding significance to our research. We have studied the comments carefully and made corrections, which we hope to meet with approval.

Best wishes,

The authors could explain a bit more why they selected Populus trichocarpa. They mention environmental aspects, but P. trichocarpa is known to have fragile branches and is not exactly suitable as a roadside windbreaker. Was there a variety / cultivar? I.e. P. lasiocarpa is known to have a generic variety and endemic var. longiamenta.

Response: Thanks for your valuable comments. The genome file and related annotation files of Populus trichocarpa can be downloaded free from the phytozome (https://jgi.doe.gov/data-and-tools/data-systems/phytozome/). Therefore, the authors used the P. trichocarpa as the woody model plant to perform the study. In addition, the studies on P. trichocarpa genome were more depth than that on another poplar, so using the genome of P. trichocarpa can help us to study comprehensively. The Nanlin 895 (Populus × euramericana CV., Nanlin 895) is a nitrogen stress-sensitive poplar clone and is a hybrid of Populus deltoides Bart. CV. × Populus euramericana (Dode) Guineir CV., developed by the Poplar Research and Development Center in China. In this study, the 'Nanlin 895' cultivar was cultured in a 1/2 MS medium for 7 days and then transferred to a 1/2 MS medium containing 0.2 M abscisic acid (ABA), 0.2 M NaCl, 2 mM H2O2, and 10% PEG6000 for abiotic stress. For drought treatment of soil-grown poplars, irrigation for 'Nanlin 895' cultivar cultivated in soil for 3 months was suspended, and leaves were harvested after two weeks. The poplars were irrigated with 0.2 M NaCl for two weeks for salt treatment.

Also, in the IJMS format, where methods are described at the end of the paper, more information about samples and methods could be integrated into the results. The reader is interested in how and from what the results came (not in detail, few words), and usually is used to the different format where this info is given above.

Response: Thanks for your valuable comments. The authors revised the results and materials accordingly.

Suggestion: all your findings are based on PCR and gene identification. Do you have access to ambient imaging techniques (DART, DESI etc.)? In your case, interesting changes might occur in stressed samples as for the position of some metabolites of interest (higher concentration closer to the surface, closer to leaf edges etc.) Worth a try to look at it in your next paper.

Response: As the reviewer pointed out, interesting changes might occur in stressed samples as to the position of some metabolites of interest. The authors will design and perform the experiments in further study according to the reviewer's suggestions.

Reviewer 2 Report

The manuscript is well written and well presented.

Many informations are provided and well discussed.

I have no special remark except few mistakes (for exemple line 46, one word seems to be missing) and some italics (Genus and species) are also missing.

Author Response

Dear Reviewer:

Thank you for the comments concerning our manuscript. The comments are all valuable and helpful for revising and improving our paper, as well as the important guiding significance to our researches. We have studied comments carefully and have made correction which we hope to meet with approval.

Best wishes,

I have no special remark except few mistakes (for exemple line 46, one word seems to be missing) and some italics (Genus and species) are also missing.

Response: Thanks for your valuable comments. The authors revised some mistakes and italics accordingly.